# Earthquake forecasting model for Albania: the area source model and the smoothing model

Edlira Xhafaj[1,2,3], Chung-Han Chan[3,4], Kuo-Fong Ma[1,2,3]

[1]Taiwan International Graduate Program (TIGP)–Earth System Science Program, Academia Sinica and National Central University, Academia Sinica, Taipei 11529, Taiwan
[2]Institute of Earth Sciences, Academia Sinica, Taipei 11529, Taiwan
[3]Earthquake-Disaster & Risk Evaluation and Management (E-DREaM) Center, National Central University, Taoyuan, 32001, Taiwan
[4]Department of Earth Sciences, National Central University, Taoyuan, 32001, Taiwan

*Correspondence to*: Chung-Han Chan (hantijun@googlemail.com)

## Abstract

We proposed earthquake forecasting models for Albania, one of the most seismogenic regions in Europe, to give an overview of seismic activity by implementing area source and smoothing approaches. The earthquake catalogue was first declustered to remove foreshocks and aftershocks when they are within the derived distance- and time-windows of mainshocks. Considering catalogue completeness, the events with M≥4.1 during the period of 1960–2006 were implemented for the learning forecast model. The forecasting is implemented into an area source model that includes 20 sub-regions and a smoothing model with a cell size of 0.2˚ x 0.2˚ to forecast the seismicity in Albania. Both models show high seismic rates along the western coastline and in the southern part of the study area, consistent with previous studies that discussed seismicity in the area and currently active regions. To further validate the forecast performance of the two models, we introduced the Molchan diagram to quantify the correlation between models and observations. The Molchan diagram suggests that the models are significantly better than a random distribution, confirming their forecasting abilities. Our results provide crucial information for subsequent research on seismic activity, such as probabilistic seismic hazard assessment.

# 1. Introduction

Albania, located in the Balkan Peninsula, belongs to the Alpine-Mediterranean seismic belt, one of the most seismic regions in Europe, often threatened by devastating earthquakes, along with Turkey and Greece (Aliaj et al., 2004; Sulstarova, 1996). High seismicity activity in the region has been the main scope of many researchers from Albania and other experts, which includes Albania as part of their seismic hazard analysis (e.g., Aliaj et al., 2010, 2004; Fundo et al., 2012; Muco et al., 2012; Shebalin et al., 1974; Slejko et al., 1999; Sulstarova, 1996), as well as multinational programs and projects within Europe, the Balkans, and the Mediterranean region (e.g., Giardini, 1999; Jimenez et al., 2003; Jiménez et al., 2001; Salic et al., 2018; Woessner et al., 2015). So far, however, no controlled research has been conducted in Albania to investigate the correlation between seismic models.

There are two primary aims of this study: (1) to investigate earthquake forecasting in Albania using different models, and (2) to assure the credibility of these models. We focus on the seismic activity considering shallow crust events, which in the Albanian case are generally at a depth of 10–20 km and, in many cases, near the surface (Sulstarova, 1996). The Albanian Seismological Network (ASN) data regarding the events from 1976 to 2000 shows that 95% of earthquakes had depths of less than 30 km (Muco, 1998; Muco et al., 2002). We investigate the seismicity of events that occurred in the region from 1960 to 2006 using the 2013 European Seismic Hazard Model (ESHM13) in the framework of the Seismic Hazard Harmonization in Europe project (referred to as SHARE), based on the SHARE European Earthquake catalogue (SHEEC). By analysing the catalogue, we aim to propose earthquake forecasting models that can be used for future research to understand the seismicity in the area and compare them with models that include an extended catalogue and seismogenic sources that are not incorporated into our forecasting model.

The time period of 46 years was chosen after the catalogue is declustered according to the Gardner and Knopoff (1974) window method to evaluate the completeness time, threshold magnitude, and Gutenberg-Richter parameters (Gutenberg and Richter, 1944) Based on the catalogue, we can forecast Albanian seismicity by implementing two models: the standard approach (Cornell, 1968) based on the area source model and the smoothing model (Frankel, 1995). Area source polygons are defined by the ESHM13, designed with the assumption that seismicity may occur anywhere within each zone,

and the delineation considers seismicity, tectonics, geology, and geodesy (Woessner et al., 2015). To avoid subjective judgments regarding how area source polygons are designed, a smoothing model is an alternative approach used to forecast seismicity. The method is based on the principle that the distribution of past events can be used to predict where future events may occur (Frankel, 1995).

Both models demonstrate a high seismic rate along the western coastline and southern part of the
60 study area, consistent with previous studies (Aliaj et al., 2004; Aliaj et al., 2010; Fundo et al., 2012) and currently active regions. To further evaluate the forecasting results from the two models, we introduced the Molchan diagram to investigate the correlation between models and observations. The catalogue from 1960 to 2006 is regarded as the "learning period" for model construction, and the seismicity during 2015–2020 is the "testing period" for comparing and validating the results. In addition, the null
hypothesis is applied to confirm the forecasting ability of the models, and the results are performed for events according to each of the threshold magnitudes, which confirms the good forecasting ability of both models. Finally, the results obtained from comparing the learning and testing periods are presented and discussed.

**2. Earthquake catalogue and analyses**

**2.1 Catalogue dataset**

To analyse the seismicity, our area of study is bounded between the latitude of 38.0°N-44.5°N and the longitude of 18.0°E-23.0°E (Fig. 1a), and a seismicity working file is created for further analysis. The SHEEC catalogue between 1900 and 2006 was compiled by the German Research Center for Geosciences (GFZ, Potsdam) and released as part of an independent project, representing a spatial-
75 temporal extract from the "European-Mediterranean Earthquake catalogue (EMEC, Grünthal et al., 2013; Grünthal & Wahlström, 2012)", which contains seismic events with moment magnitudes ranging from 3.5 to 7.0 for our region of study. We implemented events with a depth ≤ 35 km, considered shallow crustal events, according to previous studies (Muço, 1998; Slejko et al., 1999; Muco et al., 2002; Aliaj et al., 2004) and the ESHM13 (Woessner et al., 2015).

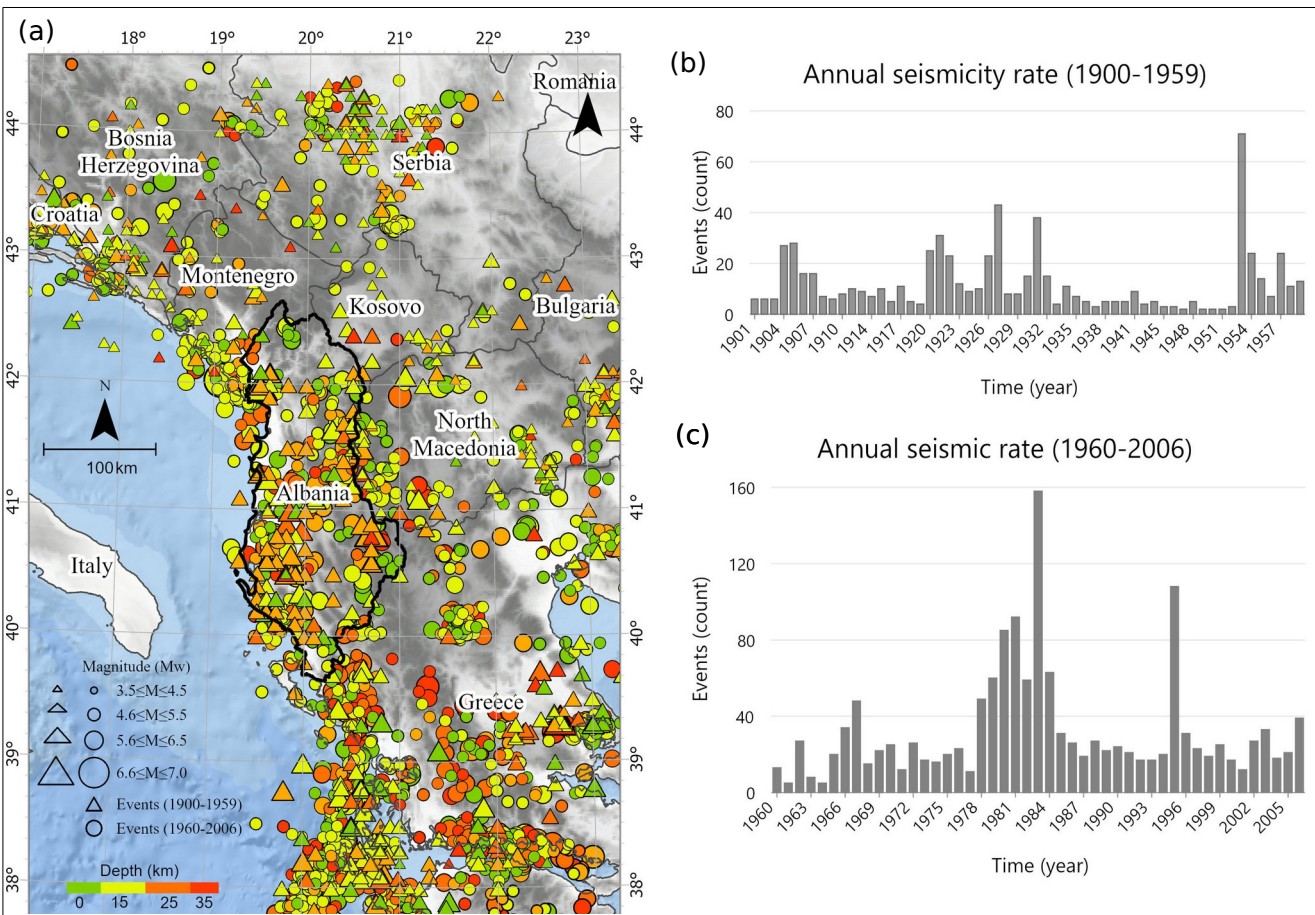

Figure 1: a) Map shows two distinct sets of seismic data related to the epicentres of shallow earthquakes (with a depth of 35 km or less) and a magnitude equal to or greater than 3.5 are represented by geometric shapes. For the first period (1900-1959), the epicentres are denoted by coloured triangles, and for the second period (1960-2006), the epicentres are represented by coloured circles. The colour of each triangle and circle corresponds to the depth of the earthquake, and the size of the circles is indicative of the earthquake's magnitude. b) The histogram corresponds to the time period from 1900 to 1959. Along the horizontal axis, time is marked in yearly increments, and along the vertical axis, the number of earthquake events is plotted. c) The histogram pertains to the time period from 1960 to 2006. Similarly, time is represented on the horizontal axis, and the number of earthquake events is shown on the vertical axis. Each bar in the histogram represents the seismic rate for a specific year within the specified time frame. The rates are calculated based on a non-declustered catalogue, meaning that all recorded earthquake events are considered without regard to potential clustering. The bars on each histogram vary in height, indicating fluctuations in the number of seismic events from year to year. Base-map provided by ESRI, plotted using ArcGISPro (Esri n.d).

The catalogue from 1900–2006 is considered to obtain completeness intervals for the entire study region using the cumulative number of events over time (Fig. 1a). When the slope changes, we consider the catalogue complete for the magnitudes above reference (Duni et al., 2010b; Markušić et al., 2016), which are also consistent with the intervals obtained from applying the (Stepp, 1972) approach.

The completeness intervals for the selected area are identified with a magnitude threshold of 4.1 for the period 1974–2006 and completed events with a magnitude of 4.5 and 5.0 after 1950 and 1901, respectively. Duni et al. (2010) and Makropoulos et al. (2012) have reported similar completeness intervals. Further analysis of this study focused on the period of time between 1960 to 2006 (Fig. 2), a period during which the catalogue is more complete and mainly based on instrumental data during the 20th century (Çağnan and Kalafat, 2012; Markušić et al., 2016).

## 2.2 Catalogue declustering

Declustering earthquake catalogue is a standard procedure for seismicity modelling to keep only the mainshocks (the largest events in an earthquake sequence) and remove events identified as foreshocks and aftershocks in a space-time window. The method is commonly used in engineering seismology and statistical seismology, e.g., probabilistic seismic hazard assessment and earthquake forecasting. A variety of techniques for declustering a catalogue to obtain background seismicity have been proposed; the majority of these methods eliminate earthquakes in a space-time window following a large occurrence known as the mainshock (Zhuang et al., 2002). The Gardner and Knopoff method (Gardner and Knopoff, 1974), also known as GK-1974, describes space-time windows dependent on the magnitude of the mainshock and denotes events inside the window of a large event such as a foreshock or aftershock. The space and time window of the GK-1974 produce a declustered catalogue that follows a Poisson distribution, which is not seen in other declustering methods (van Stiphout et al., 2012), and is given by equation (1):

$$L(km)=10^{0.1238 * M+0.983} \ , \qquad T(days)=\begin{matrix} 10^{0.5409 * M - 0.547}, if \ M < 6.5 \\ 10^{0.032 * M+2.7389}, if \ M \geq 6.5 \end{matrix}, \text{respectively,} \qquad (1)$$

where M is the magnitude of the mainshock, L is the distance from the mainshock in kilometres, and T is the time in days. Given the moment magnitude of each earthquake in our catalogue, using the algorithms from GK-1974, we calculated a specific distance L (M) and time T (M) to denote the foreshock and aftershock that took place before and after the mainshock, respectively. All the events are sorted according to their magnitudes (highest to lowest), and those events that are within the spatial and

temporal window of large events are dependent. Figure 2 (b,c) conveys information about the geographical distribution of shallow earthquakes over two distinct time periods: 1900-1959 and 1960-2006. The rates are calculated based on a non-declustered catalogue, meaning that all recorded earthquake events are considered without regard to potential clustering, offering a comprehensive view of seismic occurrences during the specified intervals. Our forecasting models are conducted using only mainshocks as presented in Fig. 2a, as considering dependent events (foreshocks and aftershocks) would lead to a higher seismicity rate (e.g., Chan, 2016).

### 2.3 The magnitude of completeness (Mc)

The magnitude of completeness (Mc) is defined as the minimum magnitude above which all earthquakes are reliably recorded, and the value varies over time and space. Mc could be estimated based on the Gutenberg-Richter Law (Gutenberg and Richter, 1944), classifying earthquakes into the number of occurrences with magnitudes greater than a given reference magnitude. The magnitude-frequency relation, the Gutenberg-Richter Law, is performed according to equation (2):

$$logN(M) = a - b * M ,\tag{2}$$

where N(M) is the number of earthquakes per year for a magnitude equal to M or larger than M, a-value (activity rate) represents the total seismic activity for a given seismic source (logN(M) for M≥0), and b-value represents the ratio between small and large events.

Identification of the completeness magnitude of an earthquake catalogue is a clear requirement for the processing of input data for seismic hazard analysis. The complete part of the declustered SHEEC is an input to estimate the spatial and magnitude probability density of seismicity in the region, the same as the approach used to obtain the seismicity density for the entire Europe (Hiemer et al., 2014).

The declustered catalogue for our area of study (as shown in Fig. 2a) is divided into 0.1 magnitude bin intervals with a minimal magnitude of 4.0 and time bins of 1.0 years starting in 1960. Figure 2b compares the annual seismic rates for shallow earthquakes for the non-declustered catalogue and the declustered catalogue, to show the impact of declustering on the seismic rates during the period from 1960 to 2006.

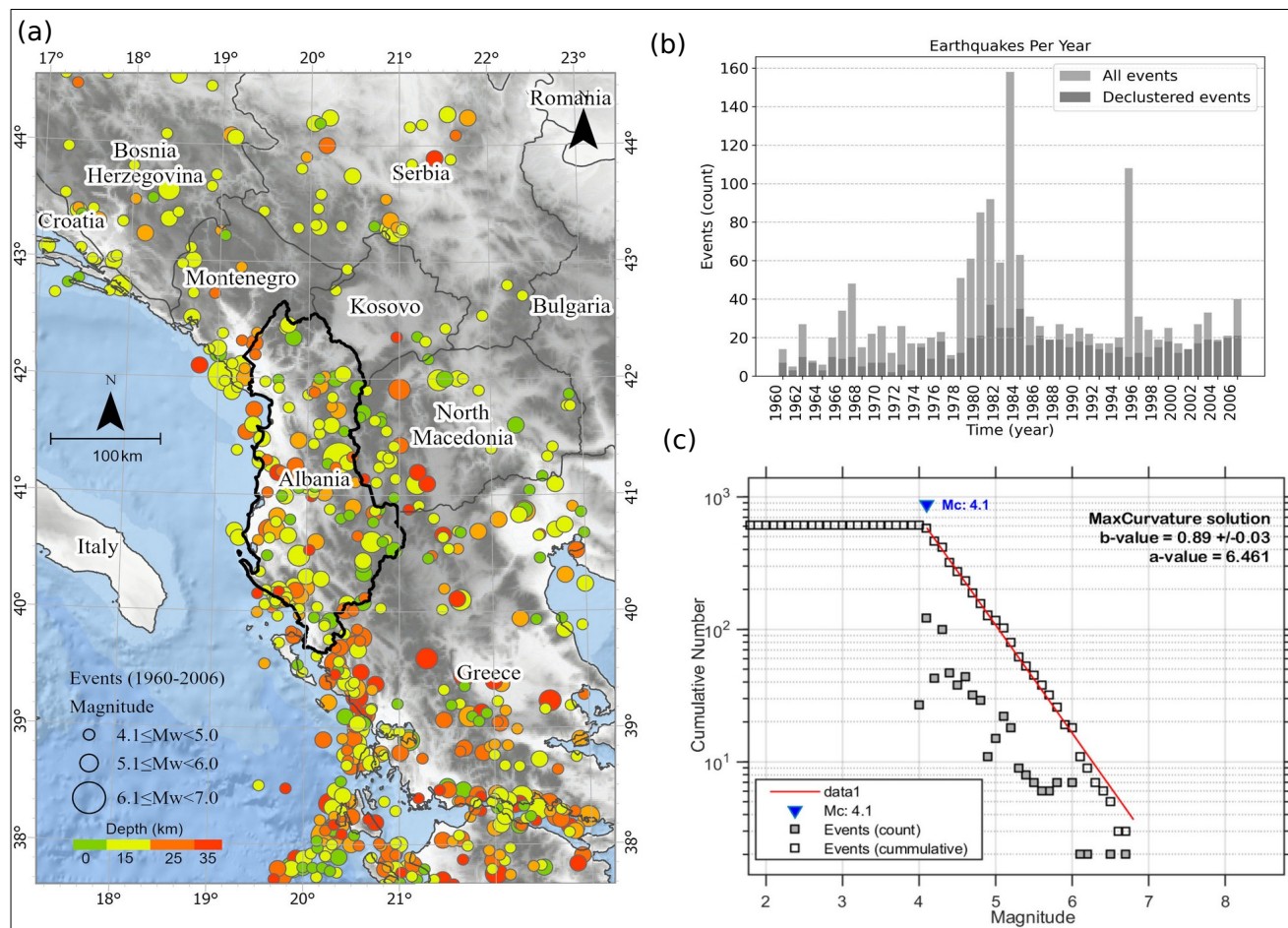

Figure 2: a) The map presents the spatial distribution of earthquake events that occurred between 1960 and 2006. Specifically, it focuses on events with a magnitude greater than 3.5, where each event is represented by a circle on the map, and the size of the circle corresponds to the magnitude of the earthquake. Additionally, a colour code is used to indicate the depth of each earthquake. Basemap provided by ESRI, plotted using ArcGISPro (Esri n.d). b) The plot consists of two histograms comparing the annual seismic rates for shallow earthquakes with a depth of 35 km or less. The light grey bars represents the non-declustered catalogue, and the dark grey bars represents the declustered catalogue. The horizontal axis denotes time in yearly increments, and the vertical axis shows the number of earthquake events. c) The empirical magnitude distribution derived from the declustered earthquake catalogue, best fit Gutenberg-Richter (1944) law obtained for the magnitude of completeness Mc = 4.1. The a and b-value obtained for the entire study region are indicated.

For our study area, the magnitude of completeness Mc = 4.1 from the Gutenberg-Richter relation was

140 obtained based on the maximum curvature method and the goodness-of-fit test on the ZMAP software (Wiemer, 2001), with an estimate of a = 5.83 and b = 0.89 value for the entire region of study (Fig. 2c).

The b-value obtained in this study is consistent with those by Grünthal et al. (2010), who reported a b-value range of 0.87 to 0.91 for a superzone covering Albania.

## 3. Earthquake forecasting models

An earthquake source model is an established approach to forecasting earthquake occurrences based on seismological, geological, tectonic, and geodetic data, with varying degrees of importance represented in the source typologies. The basic component of the forecasting model is an earthquake source model that determines the rate of earthquake activity and the rate of occurrence of events as a function of space, time, and magnitude (Hiemer et al., 2014). Here, we propose two forecasting models:
the area source and smoothing models, detailed below.

### 3.1 The area source model

        Area source models are one of the most implemented approaches to assessing seismic hazards and characterizing seismicity that occurs over large regions where single fault structure detection and classification, determination of location, geometry, and seismicity frequency parameters are difficult
(Wiemer et al., 2009). Our study area is covered by 20 area source polygons as proposed by ESHM13 (Fig. 3a), and those areas with few events have been merged into areas with similar characteristics. Seismicity activity in the form of a- and b-values (Gutenberg and Richter, 1944), the annual rate of seismic activity, and the maximum magnitude (Mmax) are evaluated for each of the area sources as given in Table 1. The table serves as a comprehensive reference for the area source parameters and
corresponding seismicity rates for the specified region of study, and can be referred  to understand the characteristics of each area in terms of seismic activity based on the provided parameters.

Table 1: The table is a reference for the area source parameters and corresponding seismicity rates for each area, the data are visualized in Figure 3. Area IDAS are consistent with those provided by the European Seismic Hazard Model 2013 (ESHM13). Additionally, the IDAS of the area with more events is retained over other merged areas. Area ID indicates the numerical references assigned to each area source as seen in Figure 3.

| ID | IDAS | TECTONICS | No. Events | Area (km2) | a | b | Mmax (Inferred) |
|----|------|-----------|-----------|-----------|---|---|-----------------|
| 1 | ALAS179 | Active Shallow Crust | 41 | 15062.45 | 4.99 (±0.075) | 0.89 (± 0.03) | 6.3 |
| 2 | MKAS180 | Active Shallow Crust | 30 | 7682.46 | 4.82 (±0.097) | | 6.9 |

| | | | | | | | |
|---|---|---|---|---|---|---|---|
| 3 | YUAS184 | Active Shallow Crust | 22 | 17080.75 | 4.52 (±0.125) | | 5.9 |
| 4 | MKAS187 | Active Shallow Crust | 23 | 15883.98 | 4.47 (±0.139) | | 6.2 |
| 5 | BAAS191 | Active Shallow Crust | 54 | 22471.91 | 4.96 (±0.076) | | 5.7 |
| 6 | BAAS192 | Active Shallow Crust | 70 | 72463.77 | 5.04 (±0.073) | | 6 |
| 7 | ITAS312 | Active Shallow Crust | 10 | 128205.13 | 4.16 (±0.176) | | 4.8 |
| 8 | GRAS369 | Active Shallow Crust | 108 | 27108.43 | 5.50 (±0.045) | | 6.6 |
| 9 | GRAS370 | Active Shallow Crust | 20 | 4437.54 | 4.71 (±0.103) | | 6.2 |
| 10 | GRAS375 | Active Shallow Crust | 20 | 10204.08 | 4.77 (±0.082) | | 5.9 |
| 11 | GRAS384 | Active Shallow Crust | 21 | 5844.7 | 5.37 (±0.052) | | 6.7 |
| 12 | GRAS385 | Active Shallow Crust | 10 | 17123.29 | 4.47 (±0.125) | | 6.2 |
| 13 | GRAS386 | Active Shallow Crust | 21 | 8267.72 | 4.79 (±0.079) | | 6.2 |
| 14 | GRAS387 | Active Shallow Crust | 21 | 22604.95 | 4.82 (±0.090) | | 6.7 |
| 15 | GRAS388 | Active Shallow Crust | 19 | 17304.19 | 4.74 (±0.090) | | 6.3 |
| 16 | HRAS995 | Active Shallow Crust | 41 | 17998.24 | 4.92 (±0.078) | | 6.9 |
| 17 | ALAS993 | Active Shallow Crust | 37 | 19151.14 | 4.82 (±0.090) | | 5.9 |
| 18 | ALAS992 | Active Shallow Crust | 59 | 24614.1 | 5.17 (±0.062) | | 6.7 |
| 19 | YUAS990 | Active Shallow Crust | 15 | 42372.88 | 4.91 (±0.055) | | 6.4 |
| 20 | GRAS371 | Active Shallow Crust | 68 | 17694.51 | 5.06 (±0.055) | | 7 |

Since there is an insufficient number of events in some areas to obtain reliable Gutenberg-Richter parameters, we considered a fixed b = 0.89 for the entire region (Fig. 2c), which is used to define the a-value for each of the areas. A uniform b-value for all the area sources is sometimes implemented by probabilistic seismic hazard assessment to minimize the effect of zonation and a low number of events inside each individual area (e.g., Chan et al., 2020; Fujiwara et al., 2013). The a-value, which represents the overall activity of the seismic source, is calculated based on the unified b-value (Table 1). The annual rate for each area source is estimated to forecast the number of events with different magnitudes within each of them, and the seismicity rate is expressed per square kilometre (Fig. 3a). Figure 3a provides a comparative view of the density seismicity rates for different area sources

within the study region; it allows for the identification of areas with higher or lower seismicity rates and may serve as a basis for understanding and predicting seismic activity across the studied region.

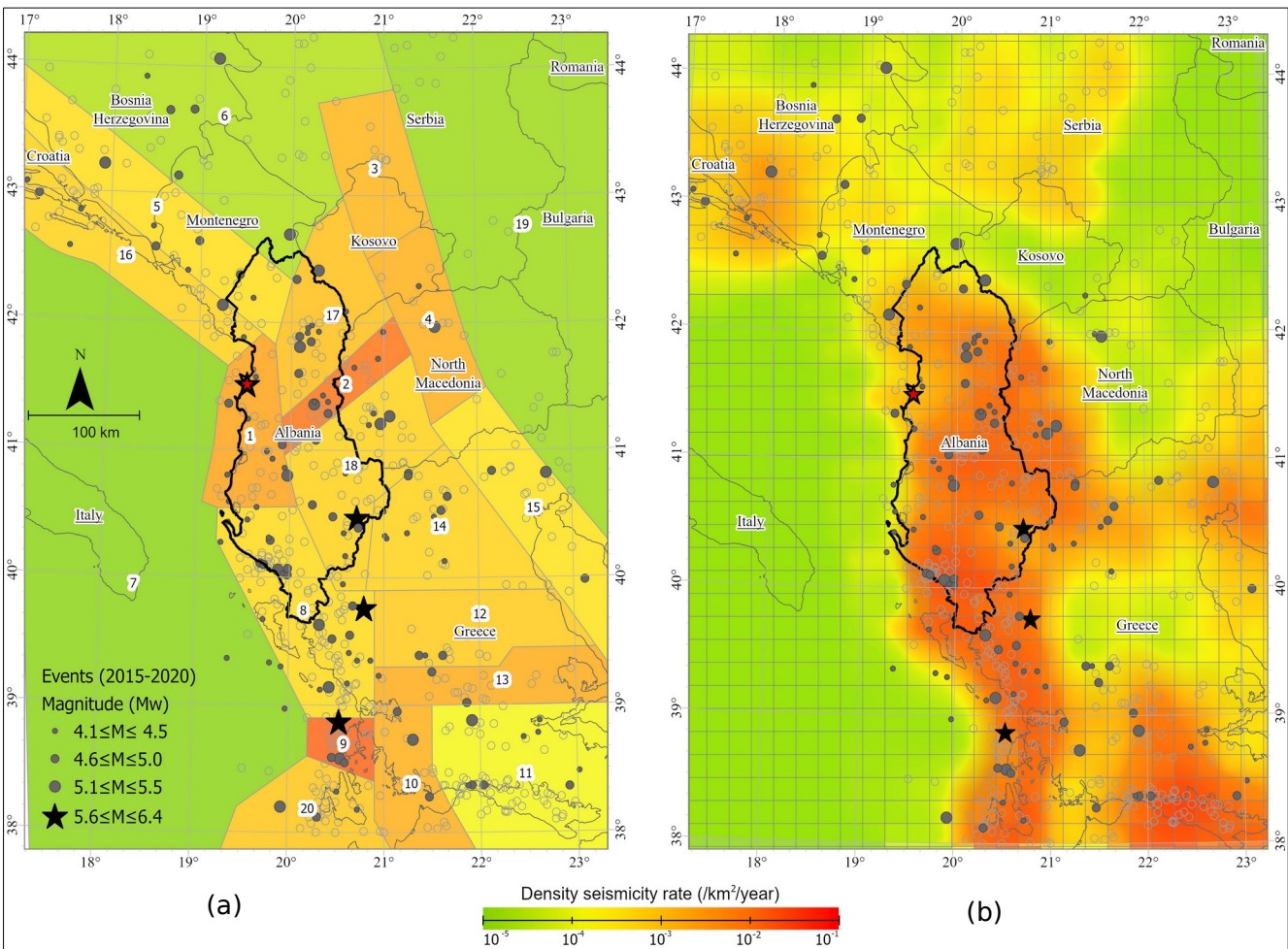

Figure 3: Density seismicity rate for the period 1960-2006 evaluated for: a) the area source model (Cornell, 1968), and b) the smoothing model (Frankel, 1995). Stars denote events with magnitude larger than 5.5, and grey filled circles with various sizes denote events of different magnitudes that occurred during the "testing period" (2015-2020) from the IGS catalogue. Grey open circles in the background represent events from the "learning period" (1960-2006) obtained from SHEEC. Each area source is indicated with an ID number, used in identifying and referencing specific parameters for each area in Table 1, while the red star is used to highlight the 2019 Mw6.4, Durres earthquake. Basemap provided by ESRI, plotted using ArcGISPro (Esri n.d).

The maximum magnitude (Mmax) for each area was estimated from the maximum observed magnitude in the catalogue using the method proposed by Kijko & Sellevoll (1992) and Fundo et al. (2012). As shown in Table 1, the area source GRAS371 (ID20) has the largest maximum magnitude in the catalogue, with a Mmax of 7.0. Duni et al. (2010), for the area including the territory of Albania,

concluded that the maximum magnitude was Mmax = 7.2 and Mmax = 6.9 for the historical and instrumental periods, respectively.

## 3.2 The smoothing model

Besides the area source model, another seismogenic source model based on the smoothing kernel, as proposed by Frankel (1995), is used for earthquake forecasting. The same approach is used to obtain the smoothed seismicity rates for the Harmonization of Seismic Hazard Maps in the Western Balkan Countries Project – BSHAP (Salic et al., 2018). The method applies a simple isotropic Gaussian smoothing kernel to derive the expected rate of events at each cell from the observed rate of seismicity in a grid of cells with a correlation distance c, represented as:

$$\widetilde{n}_i = \frac{\sum n_j e^{d_{ij}^2/c^2}}{\sum e^{d_{ij}^2/c^2}} \tag{3}$$

where $\widetilde{n}_i$ is the expected rate of events at each cell, $n_j$ is the observed rate of seismicity in a grid of j cells, $d^{ij}$ is the distance between the $i^{th}$ and $j^{th}$ cells, and c is the correlation distance for the adaptive kernel, that indicates the bandwidth parameter of the Gaussian function that controls how rapidly the kernel's weights (seismicity) diminish with distance from its center (number of events concentrated within a 0.2˚x 0.2˚ grid cell). Input parameters are the grid extend and grid cell size, the uniform b-value, bandwidth (in kilometres), completeness magnitude, and completeness year. The computed result is the observed number of earthquakes in each cell and the smoothed seismicity rate.

To apply the method, the area of study is divided into grid cells with a size of 0.2˚x 0.2˚, and the rate of earthquakes ($\widetilde{n}_i$) with M≥4.1 is counted for each cell (Fig. 3b); this count represents the maximum likelihood estimate for that cell based on the method by Weichert (1980). The grid size 0.2˚x 0.2˚ is based on the events' location uncertainty as given by ESHM13 at the range of 10 to 15 km (Woessner and Monelli, 2011). To apply the smoothing model, we follow the procedure (code) in Hazard Modeller's Toolkit, an open-source library that is related to the OpenQuake-engine hazard calculation software (Weatherill et al., 2014).

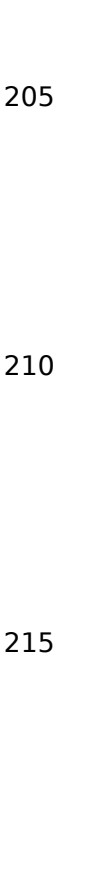
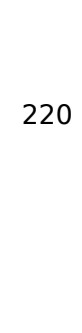
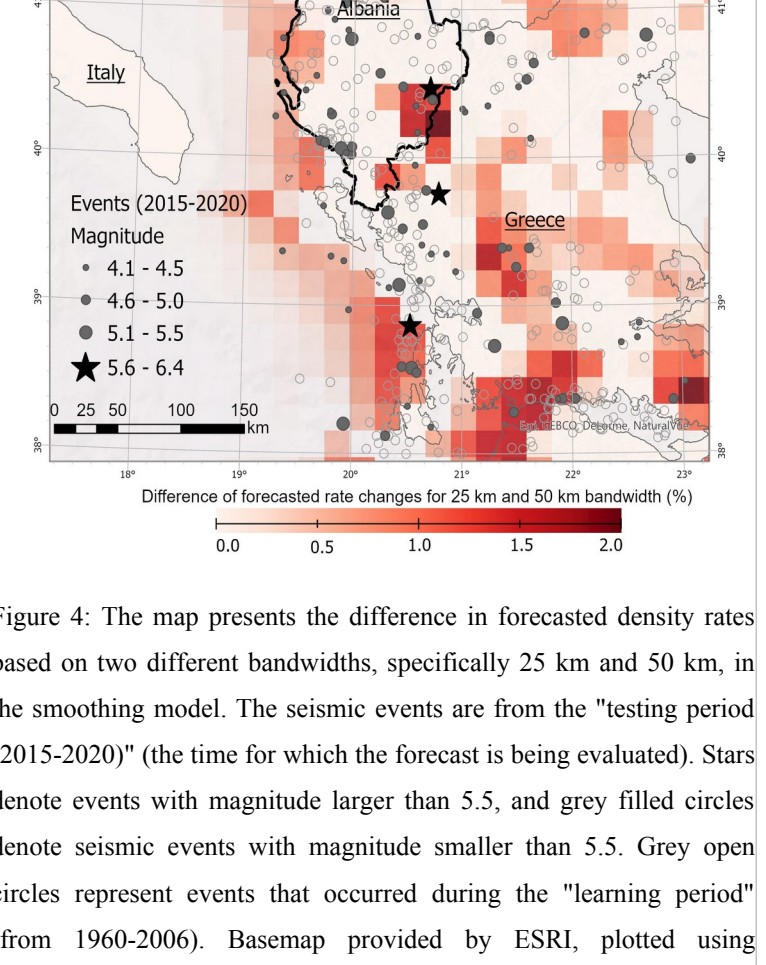

Figure 4: The map presents the difference in forecasted density rates based on two different bandwidths, specifically 25 km and 50 km, in the smoothing model. The seismic events are from the "testing period (2015-2020)" (the time for which the forecast is being evaluated). Stars denote events with magnitude larger than 5.5, and grey filled circles denote seismic events with magnitude smaller than 5.5. Grey open circles represent events that occurred during the "learning period" (from 1960-2006). Basemap provided by ESRI, plotted using ArcGISPro (Esri n.d).

The computed annual rates from the smoothed model for the bandwidth of 50 km are presented in Fig. 3b, forecasting the highest seismicity rate in the south and west of the study area, where the largest number of events is located and moderate-to-large earthquakes have occurred. In this study, the correlation distance is fixed at 50 km after testing different bandwidth values of 25 km and 50 km. As indicated in the original work by Frankel (1995), a larger than 50 km correlation distance spread out the seismicity so that details were lost, and smaller correlation distances resulted in segmented patterns of seismicity. The annual rates from the smoothed model for different bandwidth (25 km and 50 km) were obtained, and the results expressed as the difference of the forcasted density rates (in %) are presented in Fig. 4.

### 3.3 Model validation

To validate the performance of the models, the Molchan diagram approach is used (Molchan, 1990; Zechar and Jordan, 2008). This method aims to quantify forecasting ability by investigating the correlation or relationship between a model and observations of earthquake events. After obtaining the seismicity for the study region from the area source and the smoothing model, we proceed to forecast the spatial distribution of seismic events spanning the period from 2015 to 2020.

The dataset integrates catalogue and the bulletins provided by the Institute of Geo-Science of Albania, referred to as the 'IGS catalogue.' Specifically, events with magnitudes equal to or greater than 4.1 are depicted as grey dots, while events with a magnitude of 5.0 or higher are represented by black stars, as illustrated in Figure 3. The reported event's magnitude from IGS is local magnitude ($M_L$), and the conversion to moment magnitude (Mw) follows the relevant regression equations by Duni et al.
(2010a):

$$M_w = 1.624 + 0.743M_L \qquad\qquad (4)$$

One of the largest events in this period in the territory of Albania was recorded along the coastline, which occurred on November 26, 2019, with Mw6.4, the most destructive earthquake in the western part of the country. The area of study is divided into grid cells $0.2° \times 0.2°$ to obtain and validate the
250 seismicity for each of the catalogues using the area source model and smoothing model. We have defined the catalogue from SHEEC (1960–2006) as the "learning" catalogue and the IGS (2015-2020) as the "testing" catalogue. Both catalogues were declustered with the same window method by Gardner and Knopoff (1974) for shallow crustal events, as we prefer to follow similar analysis procedures for a better evaluation of our data and models. For the "testing" catalogue, we have determined the fraction
of alarm-occupied space as the percentage of observations within the region with a forecasting level equal to or higher than "alarm", and the fraction of failure in forecasting as the percentage of observations having a lower forecasting level than "alarm". Since the study region is divided into grid cells, each cell in which an earthquake is forecast to occur constitutes an alarm cell.

     A Molchan diagram plots the missing rate versus the alarming rate, and each of them gets a
260 value from 0 to 1 (0% to 100%). If the alarming rate changes from 0 to 1, the missing rate will decrease

from 1 to 0. The diagonal line from (0,1) to (1,0) would be the long-run expectation for alarms that are declared randomly, i.e., the missing rate equals the alarming rate, indicating a completely random guess.

A perfect forecast would have a value of missing alarm equal to 0 (no false alarms) and an alarm equal to 1, that is all earthquakes are perfectly forecasted (Molchan, 1990, 1991). The prediction points under the diagonal line mean the missing rate is less than the alarming rate and the prediction is better than a random guess, which is consistent with our analysis as they follow the definition given for the evaluation of source models with the Molchan diagram. We underlined that both diagrams show good performance for the targeted observations but are more suitable for large events,  as can be seen in Fig. 5a and Fig. 5b. Also, the smoothed model indicates a better forecast for future events than the area source model, as the predictive curve is always lower than the area source model's predictive curve.

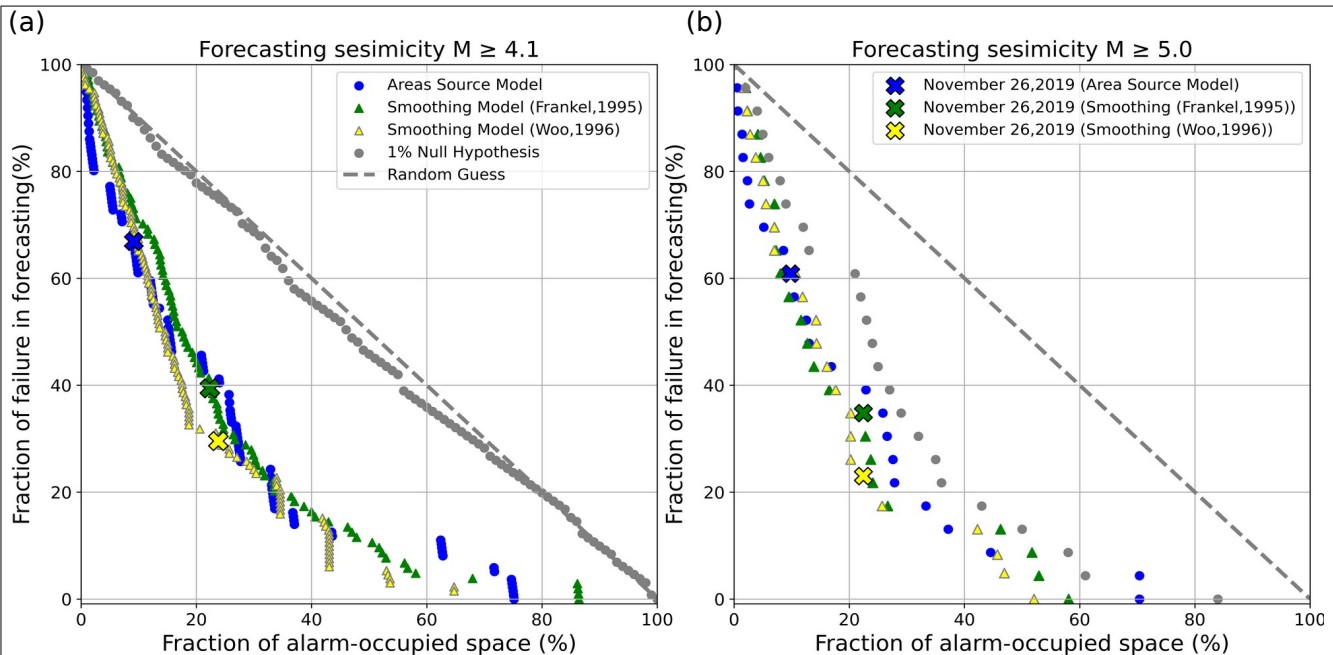

Figure 5: The Molchan diagram performance is being evaluated using the IGS catalogue for the period 2015-2020, specifically for events with different magnitude thresholds, and multiple models for comparison, as well as the significant 2019 Mw6.4 seismic event chosen to assess its impact on each model's performance. a) The diagram displays the performance of the Molchan diagram for events with a magnitude greater than or equal to 4.1. b) The Molchan diagram assesses the performance for events with a higher magnitude threshold, specifically M≥5.0. Blue dots represent the results from the area source model, green and yellow triangles show the results from two different smoothing models: one based on Frankel (1995) and the other on Woo (1996). Grey dots denote the 1% null hypothesis for 132 events with M≥4.1 and for 23 events with M≥5.0, indicating the expected distribution under the null hypothesis for comparison. Coloured crosses represent the 2019 Mw6.4 event on the diagram for each model, to distinguish the impact on each model's performance.

The forecasting performance of different source models is investigated by plotting the curve at a 99% confidence interval of the null hypothesis for the forecasting events with M≥4.1 and M≥5.0 (shown in Fig. 5a and b, respectively), confirming the good forecasting performance of the area source and smoothing model as both respective curves are under the confidence interval curve. As discussed by Schorlemmer et al. (2010), assuming a null hypothesis where the observations fall into the lower curve of the distribution, the null hypothesis is rejected.

## 4. Discussion and conclusions

The present study was designed to propose earthquake forecasting models and to discuss the seismic activity in one of the most seismic regions on the European continent, using past earthquakes to forecast future earthquakes. Two forecasting approaches are used to obtain the spatial distribution of the seismicity rate, considering events with a minimal magnitude of 4.1, which represents the threshold of catalogue completeness. The boundary is lower than the minimum magnitude (Mmin = 4.5) considered by Fundo et al. (2012) as the low bound for building damage. The annual seismicity rate for our forecasting models is determined from the complete part of the declustered earthquake catalogue, taking into account a- and b-values and the distribution of maximum magnitude (Mmax). The highest seismic activity rate is forecasted along the western coastline and southern part of the study region, which corresponds to the location of observed earthquakes as given by the earthquake catalogue, compared to the low activity rates in the central part (area source 17 and 18 for Albania, other low-density areas, refer to Fig. 3a) of the study region. The seismic rate calculated from both models, depicted in Figure 3, aligns with earlier research on seismic activity, as documented by Slejko et al. (1999), Aliaj et al. (2004), Fundo et al. (2012), Salic et al. (2018), and Woessner et al. (2015).

To evaluate the smoothing model's uncertainty and the impact of bandwidths, we compared the forecasted seismicity rates corresponding to two different bandwidths of 25 km and 50 km, which are comparable to the events' location uncertainty described in Section 3.2. The contrast in rates between the smoothing seismicity from different bandwidths reveals that variations are trivial, with an overall deviation of less than 2% across the entire study area. Furthermore, both models exhibit a high level of confidence, exceeding 98% probability, as depicted in Figure 4. Note that most of the forecast events

are in the region, with an insignificant difference in the seismicity rate. When we compare our models with observations as given by IGS, the higher seismicity rate is highlighted along the coastline (Fig. 3). The maximum magnitude based on the observed events has a value of 6.8, which is comparable to Mmax = 6.9, claimed by Duni et al. (2010) as the maximum magnitude for the instrumental period in Albania for the catalogue period from 510 BC to 2008 AD, proving that our estimations for Mmax obtained following the method proposed by Kijko & Sellevoll (1992) seem to be reasonable.

Furthermore, to test the consistency of the results from the area source and smoothing model, the credibility of our models was confirmed by the Molchan diagram, as all the events from the testing catalogue (represented by grey dots and black stars in Figs. 3 and 4) are under the diagonal line, approving the good forecasting abilities of both approaches. The models show better forecasting ability for larger events with M≥5.0 than smaller ones with M≥4.1 (Fig. 5). Many of the events occur in areas where both earthquake source models have high forecasting rates, and such a conclusion is crucial for probabilistic seismic hazard assessment. We present the location of the November 26, 2019 (Mw6.4) event (black stars, in Figs. 3 and 4) that occurred in the western part of Albania on the Molchan diagram, which appears to have a low fraction of alarm-occupied space compared to the smoothing model, confirming again a better forecasting performance compared to the forecasting performance from the area source model (Fig. 5).

The smoothing kernel approach of Frankel (1995) implemented in this study is magnitude-independent, and the spatial distribution for large magnitudes could be forecasted based on the distribution of smaller events, providing better forecasting ability. We further propose another forecasting model using a magnitude-dependent smoothing approach proposed by Woo (1996). This approach has been applied to various studies (e.g., Chan et al., 2018). The findings are graphically presented for the purpose of comparison, along with the area source and Frenkel (1995) approaches in Figure 5, revealing similar forecasting abilities between the three approaches. Findings regarding seismicity parameters and source models as presented above have significant implications for the understanding of seismic activity in our region and for raising awareness of earthquake phenomena.

Additional studies are desired for further investigation of the earthquake catalogue, including a longer period, and to integrate supplementary data regarding other seismogenic sources from geological

and tectonic information for the subsequent probabilistic seismic hazard assessment. This study can be used for future research work completed with information about fault activity, segmentation models, rupture process documentation, and seismic moment accumulation that are not incorporated into our forecasting model.

## Data availability

The data (catalogues and area polygons) in this study are provided from the European Facilities for Earthquake Hazard and Risk (EFEHR) and are available online through the the 2013 European Seismic Hazard Model (ESHM13) Overview on http://efehrcms.ethz.ch/en/Documentation/specific-hazard-models/europe/overview/. The ESHM13 was developed within the SHARE Project (the Seismic Hazard Harmonization in Europe project), and more information can be found at http://www.share-eu.org/. The SHARE European Earthquake catalogue (SHEEC) catalogue (1900-2006) was compiled by the German Research Center for Geo-Sciences (GFZ, Potsdam) and released under https://www.gfz-potsdam.de/emec/ as part of an independent project, representing a spatial-temporal extract from the "European-Mediterranean Earthquake catalogue (EMEC)". The data for the period 2015-2020 were collected by combining the catalogue and the bulletin data from the Institute of Geoscience of Albania (https://www.geo.edu.al/site/).

## Acknowledgements

We thank the editor and the reviewers whose comments helped to improve this manuscript. This study was supported by the National Science and Technology Council in Taiwan under the Grant Numbers MOST 109-2116-M-008-029-MY3, MOST 110-2124-M-002-008008, and MOST 110-2634-F-008-008. This work is financially supported by the Earthquake Disaster & Risk Evaluation and Management Center (E-DREaM) from the Featured Areas Research Center Program within the framework of the Higher Education Sprout Project by the Ministry of Education (MOE) in Taiwan. This publication is with Taiwan Earthquake Research Center (TEC) under the Contribution Number 00187.

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
