# Peer review of "Earthquake forecasting model for Albania: the area source model and the smoothing model"

_EGUsphere, 2022_

## Author Response (AR1)

**February 16th, 2023**

**To:** *Natural Hazards and Earth System Sciences*

Dear Editor,

We appreciate the constructive comments provided by you and the two reviewers and have revised our manuscript, egusphere-2022-595, entitled, "**Earthquake forecasting model for Albania: the area source model and the smoothing model**" accordingly. Below, we detailed the revision of the manuscript. All the changes are marked with underlines in the revised manuscript.

We hope the present version of the manuscript fits the standards of the journal and is now suitable for publication.

Sincerely yours,

Chung-Han Chan, PhD.

Under the major revision scope, we revised all the graphic (maps and plots) of this manuscript. The maps are product of data analysis in ArcGIS Pro (Esri, n.d.), we highlighted the source on our maps as previously the information may have been difficult to read (bottom right).

1) Figure 1: Update of the map for a better visualization of the background map source (Esri, CGIAR, USGS, Esri, HERE, Garmin, FAO, NOAA, USGS), as provided in ArcGIS Pro software automatically.

[Figure]

2) Figure 2: Update of the map for a better visualization of the background map source (Esri, CGIAR, USGS, Esri, HERE, Garmin, FAO, NOAA, USGS), as provided in ArcGIS Pro software automatically.

[Figure]

3) Figure 3: Update of the map for a better visualization of the background map source (Esri, GEBCO, DeLomre, NaturalVue), as provided in ArcGIS Pro software automatically.

[Figure]

4) Figure 5: Update of the plots for a better visualization on the understanding of data for two models.

[Figure]

Reference:

Esri. (n.d.). *2D, 3D & 4D GIS Mapping Software | ArcGIS Pro*. Retrieved February 9, 2023, from https://www.esri.com/en-us/arcgis/products/arcgis-pro/overview

No.300, Zhongda Rd., Zhongli Dist., Taoyuan City 32001, Taiwan
Tel: +886 (3) 4262242 GMT+8h
Email: chchan@ncu.edu.tw

---

## Referee Report (RR1)

In this manuscript, the authors attempt to obtain the earthquake forecasting model for Albania by area source and smoothing approaches. The study purpose is good, however, both data and result may not be sufficient. Major revision of this manuscript is required. I recommend the authors to use the English proof-reading service. Some questions and comments are listed as following:

1. The author claims that the seismicity in Albania is high, however, the 60-yrs seismicity shown in Fig. 1 does not support this statement. This is the important issue that, after decluster procedure for the M≥4.0 catalog, few events remain. Furthermore, the author divides the study area into 20 sub-regions, which means the investigated events for each sub-region is very few. For such statistical analysis, the results in the work is questionable.

2. In p.2, the author mentioned three depth ranges, which is very confusing.

3. Line 49, please list "previous studies".

4. Line 50-51, how does the author determine the time periods of learning and testing?

5. Please add a plot to show the yearly cumulative event number before and after declustering.

6. Line 155-157, I don't understand this sentence.

7. For Table 1, please also show the dimension and total event number of each ID.

8. Line 196-197, I don't understand this sentence.

9. In Figure 3, does the author set the criterion of calculation? What is the minimum number for the sufficient result?

10. Line 208, what is "completeness year"?

11. Line 214, what is "correlation distance"?

12. Line 217-218, the sentence is inconsistent with Fig. 4.

13. Fig.4 and Line 215-216, 316, the difference of forecasted rate changes for 25 km and 50 km is very small, why not use 25 km to show more detailed pattern?

14. Line 263-266, the description is not well written.

15. Line 268-270, the description is inconsistent with Fig. 5.

16. Line 274, I don't see it is more suitable for large events, since the fraction of failure is bigger than that of alarm-occupied.

17. Fig. 5, please check and correct the color and marker.
18. Line 309-311, the title is specific for "Albania", however, the analyzed area is more than Albania. So, when the author mentions some locations, I feel confused. For example, where is the "inner part"?
19. Line 312-313, please check and correct the reference style.
20. Line 317-318, I don't understand this sentence.
21. Line 325-326, I can't tell the result of M>5.0 is better than that of M>4.1.
22. Line 327-330, the description is not well written.
23. Line 328, there are two black stars in Figs. 3 and 4, which one is the November 26, 2019 (Mw6.4) event?

---

## Author Response (AR2)

**Response to Reviewer 1**

In this manuscript, the authors attempt to obtain the earthquake forecasting model for Albania by area source and smoothing approaches. The study purpose is good, however, both data and result may not be sufficient. Major revision of this manuscript is required. I recommend the authors to use the English proof-reading service. Some questions and comments are listed as following:

**Q1.** The author claims that the seismicity in Albania is high, however, the 60-yrs seismicity shown in Fig. 1 does not support this statement. This is the important issue that, after decluster procedure for the M≥4.0 catalog, few events remain. Furthermore, the author divides the study area into 20 sub-regions, which means the investigated events for each sub-region is very few. For such statistical analysis, the results in the work is questionable.

**R.** Consistent with the findings of earlier studies (Aliaj et al., 2004; Aliaj et al., 2010; Fundo et al., 2012) and when comparing seismicity in Albania to other regions in the Balkans, our analysis reveals a high level of seismicity in Albania. Also, to obtain reliable parameter for the 20 sub-regions, the uniform b-value has been considered to compute the rates for each sub-region, which has been used in similar studies, e.g., Fujiwara (2014); Chan et al. (2020), in addressing the concerns raised by the reviewer. Those clarifications are done in **L28-L33** and **L155-L157**, respectively.

**Q2.** In p.2, the author mentioned three depth ranges, which is very confusing.

**R.** Based on previous studies, most of the seismicity in our study is shallower than 35 km, that is, Sulstarova (1996) concluded the events are generally with depth at 10-20 km; Muco et al. (2002) investigated the Albanian Seismological Network (ASN) data and showd that 95% of earthquakes had depths of less than 30 km (Muco et al., 2002). Thus, we implemented the events with depth ≤ 35 km for our analysis, as shown in Figure 1 and Figure 2. We have now a more suitable rephrasing in **L36-L40** and **L75-L77**.

**Q3.** Line 49, please list "previous studies".

**R.** We now have added the references of previous studies in **L57-L59**.

**Q4.** Line 50-51, how does the author determine the time periods of learning and testing?

**R:** To ensure a reliable model, we allocated the majority of the 46-year time period for learning, setting aside the remaining 5 years to test the model's credibility. Model validation using the Molchan diagram is depicted in Figure 4.

**Q5.** Please add a plot to show the yearly cumulative event number before and after declustering.

**R:** We followed the comment to redo our plot as shown in Figure 2b.

**Q6.** Line 155-157, I don't understand this sentence.

**R:** The sentence has been rewritten as now in revised version in **L210-L215**.

**Q7.** For Table 1, please also show the dimension and total event number of each ID.

**R.** Table 1 has been updated by adding the suggested field giving the dimension and total event number of each ID.

8. Line 196-197, I don't understand this sentence.

**R:** To provide a more suitable rephrasing, we revised our sentence in **L267-L279.**

**Q9.** In Figure 3, does the author set the criterion of calculation? What is the minimum number for the sufficient result?

**R:** The minimum number to obtain reliable parameters for each area source was determined 50 events. Considering small numbers of events in several areas (as shown in Table 1), we applied a uniform b-value for the entire study region to minimize the effect of zonation and a low number of events inside each individual area (a similar procedure as Fujiwara et al., 2013; Chan et al., 2020). The description is given in **L155-L157**.

**Q10.** Line 208, what is "completeness year"?

**R:** The term "completeness year" refers to the starting time when all events, reaching the magnitude of completeness, were accurately recorded. The description is given in **L83 -L86**.

**Q11.** Line 214, what is "correlation distance"?

**R:** We have now clarified the context of "correlation distance" in **L177 -L180**.

**Q12.** Line 217-218, the sentence is inconsistent with Fig. 4.

**R:** The sentence has been revised and a detailed description of the results is given in **L293-L299**.

**Q13.** Fig.4 and Line 215-216, 316, the difference of forecasted rate changes for 25 km and 50 km is very small, why not use 25 km to show more detailed pattern?

**R:** As given in **L190-L193**, smaller correlation distances will result in segmented patterns of seismicity in the study region.

**Q14.** Line 263-266, the description is not well written.

**R:** The description has now been corrected/elaborated in **L214-L223** of the revised version.

**Q15.** Line 268-270, the description is inconsistent with Fig. 5.

**R:** We have now clarified our description in **L208-L214** on the revised version.

**Q16.** Line 274, I don't see it is more suitable for large events, since the fraction of failure is bigger than that of alarm-occupied.

**R:** To be noted in Figure 4b that the November 26, 2019 (Mw6.4) and other events have missing alarm lower than in Figure 4a, while the alarming rate moves trowards 1 in Figure 4b.

**Q17.** Fig. 5, please check and correct the color and marker.

**R:** Figure 5 has been redone and corrected in the revised version.

**Q18.** Line 309-311, the title is specific for "Albania", however, the analyzed area is more than Albania. So, when the author mentions some locations, I feel confused. For example, where is the "inner part"?

**R:** To obtain seismicity for Albania our study region has been extended up to 200 km, as described in **L69-L70**. The "inner part" clarification has been added in **L268-L272**.

**Q19.** Line 312-313, please check and correct the reference style.

**R:** The references have been check and correct following the journal style.

**Q20.** Line 317-318, I don't understand this sentence.

**R:** The sentence has been rewritten as now in revised version of **L263-L265**.

**Q21.** Line 325-326, I can't tell the result of M>5.0 is better than that of M>4.1.

**R:** By definition a perfect forecast would have a value of missing alarm equal to 0 (no false alarms) and an alarm equal to 1, that is all earthquakes are perfectly forecasted (Molchan, 1990; Molchan, 1991). Following the approach, it can be noted in Figure 4b that the November 26, 2019 (Mw6.4) has missing alarm lower than in Figure 4a, while the alarming rate moves trowards 1 in Figure 4b.

**Q22.** Line 327-330, the description is not well written.

**R:** The description has now been corrected/elaborated in **L276-L279** of the revised version.

**Q23.** Line 328, there are two black stars in Figs. 3 and 4, which one is the November 26, 2019 (Mw6.4) event?

**R:** Figure 3 has been regenerated, and the November 26, 2019 (Mw6.4) event has been denoted to be differentiated from the other events.

**Responses to Reviewer 2**

**Q.** The treatment of smoothing could be (indeed should be) much more sophisticated. Rather than simple activity rate smoothing, as implemented by Frankel in 1995, it is of interest in earthquake forecasting to consider kernel smoothing of event epicentres, as originally suggested by David Vere-Jones (1992): Statistical methods for the description and display of earthquake catalogs. This paper is a chapter in 'Statistics in the Environmental & Earth Sciences'. It was this paper by Vere-Jones, which motivated the 1996 BSSA paper by Gordon Woo, which is included in the references. There are many options for the event smoothing kernel. It need not be magnitude-dependent. Some of these options should be considered here. Earthquake forecasting is such an important and controversial subject, that the best statistical methods, (e.g. Vere-Jones' kernel event smoothing), should be applied. This can be done for Albania.

R: Thanks for the comment and the suggestion. Following the smoothing approach by Woo (1996), we obtained the smoothed seismicity rates considering the kernel smoothing of event epicenters. The results are shown in Figure 5, now added in the manuscript in **L298-L304**.